# Design and Experimental Study of a Wine Grape Covering Soil-Cleaning Machine with Wind Blowing

Qizhi Yang [1,2,*], Mingsheng He [1], Guangyu Du [1], Lei Shi [1], Xiaoqi Zhao [1], Aiping Shi [1,2] and Min Addy [3,*]

[1]  School of Agricultural Engineering, Jiangsu University, Zhenjiang 212013, China; hms5456@163.com (M.H.); duguangyu092@163.com (G.D.); 13862854636@163.com (L.S.); 18852852600@163.com (X.Z.); qzyrobot@126.com (A.S.)

[2]  China Wine Industry Technology Institute, Zhongguancun Innovator Center, Yinchuan 750021, China

[3]  Bioproducts and Biosystems Engineering Department, University of Minnesota, Saint Paul, MN 55108, USA

[*]  Correspondence: yangqz@ujs.edu.cn (Q.Y.); minxx039@umn.edu (M.A.); Tel.: +86-138-1517-2929 (Q.Y.)

**Abstract:** Due to the cold and dry climate during the winter season of Central Asia, in order to prevent frostbite and vines drying out for wine grapes, the common methods are burying the vines in winter under a thick layer of soil and then cleaning them out in the next spring. The design of existing vine digging machinery is not precise enough and can only remove the outer layer of the soil on both sides and the top. It cannot clean the soil from the central area of the buried vine. Sometimes, the branches and buds get damaged due to uneven driving condition. To solve the problem, an innovative non-contact blower was designed and tested to clean the vine. In this paper, the design specifications and operation parameters of the blower were determined according to the agronomic properties of the grapevines. Fluent-EDEM coupling, that is, with the help of Engineering discrete element method (EDEM) and CFD fluid simulation software Fluent, was the most common method for dynamic simulation of gas-solid two-phase flow. The Fluent-EDEM coupling simulation was used to simulate the dynamics of soil particles under the action of different wind speeds and blowing patterns, with the goal of a high soil cleaning rate. A prototype of the soil cleaning blower was manufactured and tested at the vineyards of Ningxia Yuquanying Farm in China. The results showed that the blower had an operation efficiency of 4669 $m^2 \cdot h^{-1}$, with an average soil removal rate of 80%. The efficiency of covering soil cleaning and rattan raising was greatly improved, and the damage rate of the vines, branches and the buds was greatly reduced.

**Keywords:** wind blowing; covering soil-cleaning machine; soil discrete particle group; dynamics simulation





## 1. Introduction

The grape industry is one of the most important industries in the world, and the grape producing areas are mainly concentrated at latitude of 30 degrees north. Kazakhstan, Uzbekistan, northwest China and other regions in Central Asia are located in the world's "golden zones" for grape cultivation. However, due to the cold and dry winter climate in these regions, vine burying in winter and vine lifting in spring must be carried out to prevent vine damage. The vine burying is done to ensure the safe overwintering of grapes, that is, to bury grapevines with soil before winter. A photo after burying the vine is shown in Figure 1. The vine lifting operation is a process of removing the covering soil and lifting the vines back to the rack in spring [1]. If not timely operated, the vines germinate in the soil at a certain temperature and humidity, which results in the branches and buds becoming damaged, and then yield decreases. However, this operation is highly labor-intensive and has low efficiency, which hinders the large-scale development of the grape industry and limits the grape growers' income. Therefore, it is very important to mechanize the soil-cleaning and vine-lifting process for the wine grapes industry to enhance the wine

grape production efficiency, so that it is possible to boost the economic and social benefits for the regional wine industry.

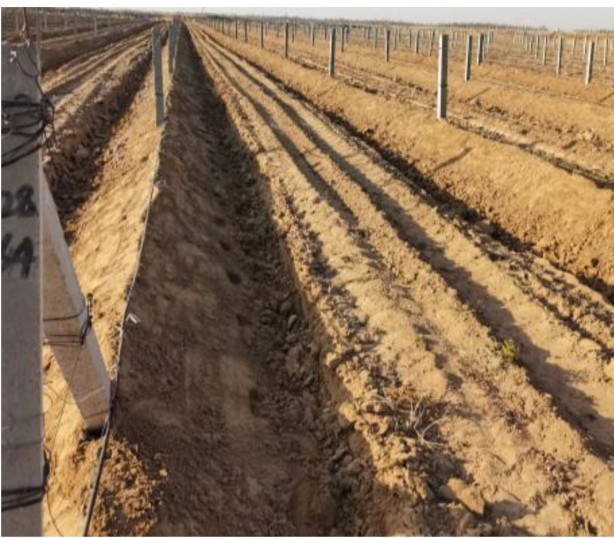

**Figure 1.** Buried grapevine pictures in November 2019.

Currently, there is little research on wine grape cleaning machines. The top grape-growing zones in European and American countries, such as Bordeaux in France, Tuscany in Italy and California in the United States, have relatively warm and humid winter climates. Therefore, there are no mandatory requirements for burying the vines in winter. However, Asian countries' grape-growing zones, such as China, Kazakhstan, Uzbekistan, etc., have cold and dry winter climates, so the research on soil-cleaning and vine-lifting machines is mainly concentrated in Asia, especially in China. In China, the research on the soil-cleaning and vine-lifting machines has been carried out for many years. For example, Liu et al. [2] designed a conical spiral vine digging machine for grapes in spring, which rotated from the inside to the outside through the spiral knife and cleaned the soil from the middle to the sides. Liu et al. developed an automatic, obstacle-avoiding, grapevine digging machine [3], which scraped and pushed the soil covering on both sides of the grapevine through the scraper, and then conveyed the soil through the auger. The machine can realize real-time pile avoidance through the hydraulic device. Zhou developed a grapevine-lifting machine [4], which adopted a gantry frame structure across the vine ridge to avoid the cement pillar in the middle and carried out the compound soil cleaning operations through the bulldozing board, brush and lifting mechanism. However, there is no research on the non-contact method of cleaning soil and lifting vines. The pneumatic method could effectively realize the mechanical methods of grasping and pushing in many cases. The seeds and shoots of crops were relatively fragile. Therefore, the pneumatic non-contact method has been gradually applied in the field of agricultural equipment, especially having many applications in the sowing and metering links. For example, Yang of Huazhong Agricultural University studied the seed discharge device of an air-suction potato seeder [5], which achieved precise seeding of potato tuber seeds by air suction. Lu et al. from Heilongjiang Bayi Agricultural University developed a potato pneumatic precision seeder [6], which adopted the suction seeding method of negative pressure suction and positive pressure blowing to achieve precise and high-speed potato seeding. Zulin and Karayel et al. [7,8] studied the impact of air flow velocity, linear velocity and other related factors on the performance of the seed metering device by improving the hydro-pneumatic seed discharge device. Karayel and Barut et al. [9,10] studied the impact of seeder speed and seeding depth on a pneumatic seed discharge device and established related mathematical models.

In summary, most of the existing grape soil cleaners suitable for wine grape production in Central Asia use scrapers to scrape or sweep off the covering soil of grapevines with a cleaning device, that is, contact soil cleaning, which not only easily damages the branches and buds of the grapevines, but also cannot remove all the soil covering in the core area of the vines. At the same time, the application of pneumatic methods in agricultural equipment, such as a seed metering device and planter, is feasible, but because of its relatively small working space, there are only some basic studies at present and the use of a blower to clean and lift the vine has not been studied, nor have the complex dynamics of soil particles under high-pressure airflow been reported.

To solve the above problems, this paper designed and developed a non-contact blower for wine grape cleaning, and the key design parameters and operating specifications were determined through simulation by Fluent-EDEM coupling, which was the most common method for dynamic simulation of gas-solid two-phase flow.

## 2. Materials and Methods

### 2.1. Agronomic Requirements

Due to the harsh field operating conditions in the Central China area, environmental factors greatly affect the operating efficiency and effectiveness of agricultural machinery. For the pneumatic soil removal equipment, the air flow needs to be precisely controlled to prevent nutrient-containing soil from being blown away from the planting area. Certain technical measures, including studying the gas–solid two-phase flow pattern, need to be considered to reduce the soil dispersion and improve the soil cleaning efficiency.

According to the agronomic requirements for grape planting and field spacing of the vineyard, the main technical parameters of the soil-cleaning and vine-lifting machinery were determined, and the following parameters were optimized by calculation, for example, electric fan power, the fan speed and the air speed range, as shown in Table 1.

**Table 1.** Technical parameters of the machine.

| Technical Parameters of the Machine | Parameter Value |
| --- | --- |
| Chassis width (working along the axis between ridges) | ≤1800 mm |
| Horizontal extension length of airduct (from chassis axis) | ≤1650 mm |
| Length of the whole machine | ≤2000 mm |
| Length of operating system (including tractor) | ≤6000 mm |
| Machine advancement speed | $\geq$0.4 m·s$^{-1}$ |
| Electric fan power | 3–20 kW |
| Fan speed | 2500–3500 r·min$^{-1}$ |
| The fan impeller diameter | 500 mm |
| The airduct section at the end of the airduct | 275 × 180 mm |
| The air speed range | 15–25 m·s$^{-1}$ |

### 2.2. The Schematic Design of the Wind-Blown Soil Cleaner

As shown in Figure 2, the soil cleaner machine included three major components: a chassis (in grey), a fan (in blue) and a power device (in green). The chassis was used to carry all the devices, and it can be installed on the rear of the tractor through the front bracket using suspension, semi-suspension or dragging, and the bottom was installed with limited-depth wheels or support wheels.

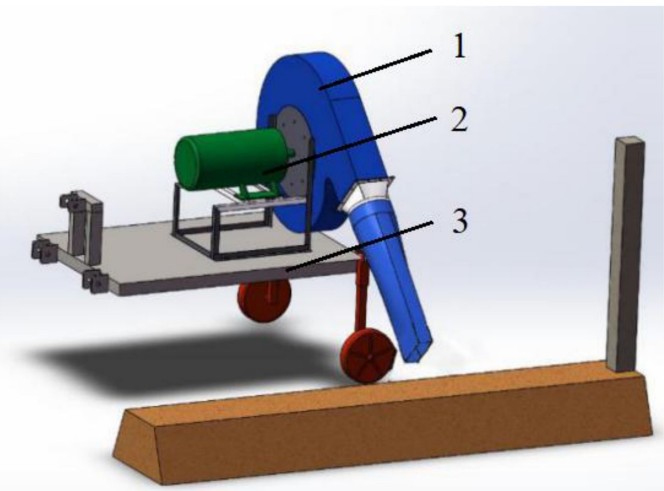

**Figure 2.** Overall structure. 1: fan, 2: power device, 3: chassis.

### 2.2.1. Design of Wind-Blowing Device

The blower was the key component of the soil-cleaning machine, including a fan and a windpipe. The soil-cleaning machine achieved the purpose of soil cleaning by shearing off the soil ridge. The air pressure of the centrifugal fan needs to be high. The fan was arranged on the longitudinal axis of the chassis to enhance the stability of the whole machine. The pointing angle of the windpipe and the distance from the soil ridge were obtained through simulation and experiment.

As the air pressure that can truly destroy the soil ridge was the dynamic pressure, according to Formula (1), the dynamic pressure was related to the density and flow velocity of the air. In the windpipe, the air was basically not compressed, so the parameter that determined the dynamic pressure was only the velocity of air.

$$p = \frac{1}{2}\rho v^2 \tag{1}$$

Since the test prototype needed to be tested for the angle of the windpipe, the air outlet of the fan and the air inlet of the windpipe were connected through a transfer tube and a hose.

### 2.2.2. Power Selection

During the development period, a 380 V fan motor was used to power the fan. In the subsequent practical applications, it was planned to use a gasoline engine to drive the fan and generate wind power by belt.

### 2.3. Determination of EDEM Simulation Model Parameters

In order to obtain the detailed design parameters of the prototype, it was necessary to study the dynamics of the soil discrete particle group under high-pressure wind blowing, the movement law and the interaction mechanism of the soil particle group, and then obtain the design parameters under the ideal soil particle dispersion state, such as wind pressure, wind speed, angle of action, duct structure and wind flow pattern, etc. The dynamic analysis of discrete particle groups was a difficult point, and there is no mature theoretical research yet. Currently, EDEM simulation analysis of the dynamics of discrete particle groups is one of the feasible methods.

EDEM are widely used in the field of agricultural engineering, such as the design and optimization process of farming, seeding, fertilization, cleaning and other related machinery. Through the coupling simulation of EDEM and CFD software, the solid–gas system can be simulated for particles to analyze the particle–particle, particle–wall and

particle–fluid interactions. CFD was computational fluid dynamics, which used the fast computing power of computers to obtain approximate solutions to fluid control equations. Fluent-EDEM coupling [11] is the most common method for dynamic simulation of gas–solid two-phase flow [12,13], since the fluid itself does not have a fixed shape, and it is difficult to observe and measure. When a large number of particles collided with each other, the movement and force of each particle were different. The fluid and the particles would also interact with each other and form a strong coupling effect, which increased the complexity of the simulation. Therefore, it was particularly important to calculate the experimental and simulation parameters of particle parameters.

### 2.3.1. Acquisition of Soil Parameters

The soil physical parameters necessary for EDEM modeling need to be obtained, including moisture content, density and particle size, and the physical property parameters including elastic modulus, shear modulus and Poisson ratio [14].

### Measurement of Soil Physical Parameters

In this section, Ningxia, one of the main wine grape-producing areas in Northwest China, has been taken as an example. According to the international norm of "Soil. Determination of dry matter and water content. Gravimetric method", standard number HJ 613-2011, the soil moisture content was measured by drying samples in an oven at temperature of 105 °C for 72 h. Samples were taken out after drying, and the moisture content of the sample was measured, as shown in Table 2.

**Table 2.** Content of the soil.

| Number | Aluminum Box Weight/(g) | Original Soil Weight/(g) | Dry Soil Weight/(g) | Water Weight/(g) | Moisture Content/(%) |
|---|---|---|---|---|---|
| 1 | 11.05 | 41.39 | 37.78 | 3.61 | 8.70 |
| 2 | 11.23 | 40.52 | 37.88 | 3.29 | 8.10 |
| 3 | 10.62 | 41.34 | 36.14 | 5.20 | 12.58 |
| 4 | 11.46 | 40.87 | 35.92 | 4.95 | 12.11 |
| 5 | 10.85 | 40.89 | 37.65 | 3.24 | 7.90 |
| 6 | 10.62 | 40.77 | 37.55 | 3.22 | 7.90 |
| average value | 10.97 | 40.96 | 37.15 | 3.92 | 9.55 |

For the soil in the core area of the vine edge after mechanical cleaning, the third and fourth groups had 5–10 cm soil, because water evaporation was not as much as 0–5 cm from surface soil (Groups 1 and 2), and there was no bottom soil (Groups 5 and 6), so the moisture content was higher.

According to the agricultural norm of "cutting ring method for determination of field water-holding capacity in soil", standard number NY/T 1121.22–2010, the soil sample was cut with a soil sampling ring-knife, and the mass of the ring-knife was subtracted after weighing to get the mass of the soil sample. The volume of the ring knife was the volume of the soil sample, and then the soil density of the sample was calculated. The calculated results are shown in Table 3.

**Table 3.** Soil density.

| Soil Depth/(cm) | Soil Density/(g·cm$^{-3}$) |
|---|---|
| 0–5 cm | 1.655 |
| 5–10 cm | 1.680 |
| 10–30 cm | 1.645 |

Calculation of Soil Physical Parameters

The elastic modulus of the soil was obtained by triaxial compression test, and the method of continuous loading and unloading was adopted for the soil sample. In this process, the axial deformation of the sample was determined, and the elastic modulus of the soil was calculated according to the international norm of "Standard Method of Test for Determining the Resilient Modulus of Soils and Aggregate Materials", standard number AASHTO T 307-1999. After measurement and calculation, the elastic modulus of the soil [15] was $50 \pm 1.37$ kPa.

According to the international norm of "Testing of soil-determination of deformation modulus by circular plate load test", standard number JUS U.B1.047-1997, in practical applications, the value can be estimated according to the Poisson's ratio Formula (2). In this paper, the static friction angle, $\varphi$, was $30.5°$. According to Formulas (2) and (3), the soil Poisson's ratio, v, was calculated as 0.35.

$$nu(v) = \frac{K_0}{1 + K_0} \qquad (2)$$

$$K_0 = 1 - \sin\varphi \qquad (3)$$

where nu (v) is the Poisson's ratio of soil, $K_0$ is the soil lateral pressure coefficient and $\varphi$ is the effective static friction angle of soil.

According to Formula (4), the calculated shear modulus was $18.52$ MN·m$^{-2}$:

$$G = \frac{E}{2(1 + nu(v))} \qquad (4)$$

where G is the soil shear modulus and E is the soil elastic modulus.

Analysis and Modeling of Soil Particle Size

According to the field conditions, all the soil in a 0.2 m long soil ridge was used for soil particle size analysis. The total mass of this part of the soil was about 47 kg. By screening the soil into different particle sizes, the distribution of the soil was obtained, as shown in Table 4.

**Table 4.** Soil particle size.

| Parameter | Particle Size Name | | | |
|---|---|---|---|---|
| | **Clod** | **10 mm Aggregate** | **5 mm Aggregate** | **Fine Sand** |
| Particle quality/(kg) | 4.8 | 9.2 | 20.7 | 12.3 |
| Percentage of particles/(%) | 10.2 | 19.6 | 44 | 26.2 |

Note: The clod referred to soil particles larger than 10 mm.

As shown in Figure 3a, the parameters obtained above were used to establish the soil particle model in EDEM software for the soil ridges. According to the mass ratio, the corresponding physical and physical property parameters were input to create four simplified soil particle models in the particle factory, as shown in Figure 3 below [16].

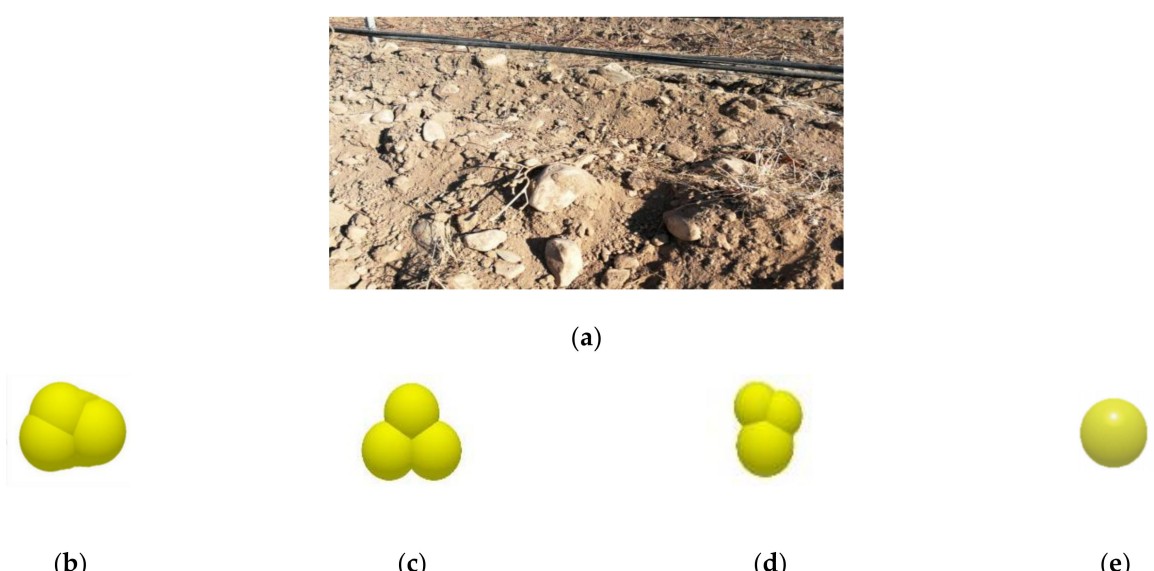

**Figure 3.** Soil particle models. (**a**) Field soil, (**b**) Ellipsoid, (**c**) 10 mm aggregate, (**d**) 5 mm aggregate, (**e**) granular.

### 2.3.2. Determination of Simulation Parameters

Selection and Calculation of Fan

According to the requirements of the project, the area of the vineyards to be processed in an hour was 4669 m². Take the vineyards of Yuquanying Farm in Ningxia province, China, as an example, where the average row spacing of the vineyards was 3.1 m. The moving speed of the soil cleaner can be obtained without considering the lane change during the operation of the soil cleaner.

$$
\begin{aligned}
v \; &= \frac{4669 \text{ m}^2 \cdot \text{h}^{-1}}{3.1 \text{ m}} \\
&= 1506.13 \text{ m} \cdot \text{h}^{-1} \\
&= 0.42 \text{ m} \cdot \text{s}^{-1}
\end{aligned}
\tag{5}
$$

The cross-sectional area of the soil ridge in the core area of the vine edge was trapezoidal, with an upper bottom of 0.35 m, a lower bottom of 0.6 m, and a height of 0.3 m. The volume (V) and mass (m) of the soil that the soil cleaner needs to remove per second is as follows, namely:

$$
\begin{aligned}
\frac{dV}{dt} &= (0.35 \text{ m} + 0.6 \text{ m}) \times 0.3 \text{ m} \times 0.5 \times 1 \text{ s} \times 0.42 \text{ m} \cdot \text{s}^{-1} \\
&= 0.05985 \text{ m}^3 \cdot \text{s}^{-1}
\end{aligned}
\tag{6}
$$

$$
\begin{aligned}
\frac{dm}{dt} m &= \rho_1 \times V \\
&= 149.625 \text{ kg} \cdot \text{s}^{-1}
\end{aligned}
\tag{7}
$$

where $\rho_1$ = 2500 kg·m$^{-3}$.

Since the internal binding force of the soil was mostly generated by its own weight, the internal force between the soil was not considered temporarily, and the soil ridge was regarded as a whole. If the soil was effectively cleaned until the vines were exposed, the covering soil ridge was pushed horizontally along the width, s, at least 0.6 m. The force required to push the soil ridge was simplified to $F_1 = \mu mg$, and the work required to achieve the above effect was $W_1$. The kinetic energy conversion method was as follows:

$$
\begin{aligned}
F_1 &= \mu mg \\
&= 972.56 \text{ N}
\end{aligned}
\tag{8}
$$

$$W_1 = F_1 \times s$$
$$= 583.54 \text{ J} \tag{9}$$

where $\mu = 0.65$ is the static friction coefficient between soil particles when the soil moisture content was maximum according to the calibration of soil parameters in Ningxia (the doi (Digital Object Identifier) was 10.11975/j.issn.1002-6819.2020.01.005 [17]).

Regardless of the wind loss after leaving the fan, it can be approximately considered that the kinetic energy of the wind was all converted into $W_2$, which was calculated as follows:

$$W_2 = \tfrac{1}{2} \times m_0 \times v_0^2$$
$$= \tfrac{1}{2} \times \rho_2 \times Q \times 1 \times v_0^2 \tag{10}$$
$$= \tfrac{1}{2} \times \rho_2 \times S \times 1 \times v_0 \times v_0^2$$

$$W_1 = W_2 \tag{11}$$

$$v_0 = \sqrt[3]{\frac{2 \times W_2}{\rho_2 \times S}} \tag{12}$$

$$P = F_1 \times v_0$$
$$= 18.7 \text{ kW} \tag{13}$$

where $\rho_2 = 1.29 \text{ kg·m}^{-3}$.

Among them, the air volume is $Q = v_0 \times S$, $v_0$ is the air outlet speed of the fan duct, S is the cross-sectional area of the air outlet, $S = 0.42 \text{ m·s}^{-1} \times 0.3 \text{ m} \times 1 \text{ s} = 0.126 \text{ m}^2$, the solution is $v_0 = 19.29 \text{ m·s}^{-1}$ and the required fan air volume Q was calculated as $5905.87 \text{ m}^3\text{·h}^{-1}$.

According to the results, considering the wind power loss and the time consumed by the soil cleaner in lane changing, the principle prototype used a 22 kW high-pressure centrifugal fan. The main parameters of the fan were as follows:

Power: 22 kW, speed: 2900 r·min$^{-1}$, weight: 200 kg, flow: 6032–9500 m$^3$·h$^{-1}$, total pressure: 6527–7610 Pa, air outlet size: 275 × 180 mm, overall size: 1223 × 1083 × 832 mm.

Selection of Power Source

(1)    Power selection

The principle prototype used an electric fan with a motor-rated power of 22 kW, so a 380 V industrial alternating current was used to power the motor during the field test. In the field test at the vineyard, a diesel generator was used to supply power to the electric motor. The diesel generator with a rated power of 80 kW was selected based on the generator efficiency of 30%.

(2)    Cable selection

$$I = \frac{P}{\sqrt{3} \cdot U \cdot \cos \varphi}$$
$$= \frac{22000 \text{ W}}{\sqrt{3} \times 380 \text{ V} \times 0.8} \tag{14}$$
$$= 41.78 \text{ A}$$

Finally, a 10 mm$^2$ four-core copper cable was selected, and its length was 30 m according to the selection principle of electrical cables [18], its doi was 10.16589/j.cnki.cn11-3571/tn.2015.03.088.

Determination of Duct Size

The design principle of the airduct was to ensure that the dynamic pressure of the air outlet was sufficient, and the area of the air outlet was as large as possible to ensure the efficiency of soil cleaning. The flow rate of the fan of the principle prototype was Q1 = 6032–9500 m$^3$·h$^{-1}$ and the wind speed at the air outlet of the fan could be calculated by Formula (15), which was the wind speed at the airduct inlet. It could be seen that under the constant flow rate, the smaller the cross-sectional area of the duct, the greater the wind speed. Fluent software was used to perform fluid simulation of the air in the pipe [19], and

the variation of wind speed in the air pipe was simulated. Thus, the outlet velocity could be obtained, and the dynamic pressure of the outlet was calculated by Formula (16).

$$v = \frac{Q}{A} \tag{15}$$

$$P = \frac{1}{2}\rho v^2 \tag{16}$$

In Formula (15), A is the cross-sectional area of the pipe. The principle prototype adopted a high-pressure centrifugal fan: the fan outlet size was $275 \times 180$ mm, and the outlet wind speed, $v_1$, was about 68 m·s$^{-1}$. According to the operating conditions, the pipe length was 500 mm. Because the fan outlet was rectangular, the airduct inlet was also rectangular. Therefore, the shape of the airduct mainly consisted of square pipe and a pipe coupling. The fluid simulation analysis of two pipes was carried out respectively [20].

### 2.4. Square Tube Fluid Simulation

Assuming that there was no turbulence and no flow deviation from the direction parallel to the airduct axis in the internal flow field of the fan, the fan impeller had no effect on the airflow. The analysis was shown in Figures 4–6:

(1) Both air inlet and outlet were $2.15 \times 10^2 \times 1.8 \times 10^2$ mm rectangular tubes
(2) Inlet $2.15 \times 102 \times 1.8 \times 102$ mm, outlet $1.8 \times 102 \times 1.8 \times 102$ mm tapered square tube
(3) Imported $2.15 \times 10^2 \times 1.8 \times 10^2$ mm, outlet $1.6 \times 10^2 \times 1.6 \times 10^2$ mm tapered square tube

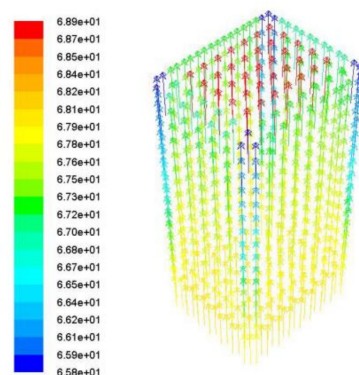

**Figure 4.** Simulation of $2.15 \times 10^2 \times 1.8 \times 10^2$ mm airduct.

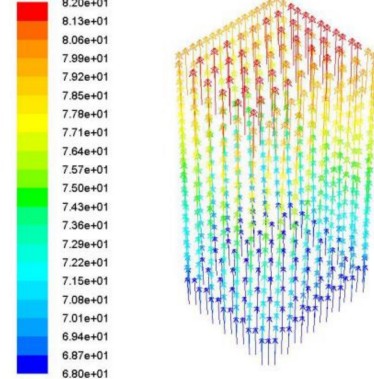

**Figure 5.** Simulation of the $1.8 \times 10^2 \times 1.8 \times 10^2$ mm airduct.

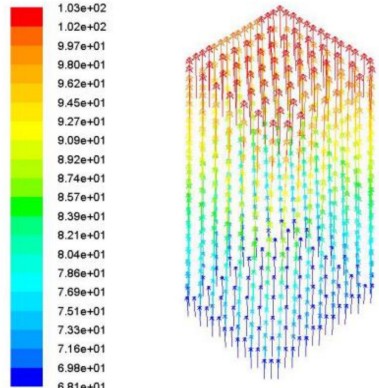

**Figure 6.** Simulation of $1.6 \times 10^2 \times 1.6 \times 10^2$ mm airduct.

### 2.5. Pipe Coupling Fluid Simulation

According to the simulation results of the square pipe, a pipe coupling was designed with square inlet and round outlet. The diameter of the round end of the air outlet should not be larger than the length or width of the square end of the air inlet, that is, the projection on the round end in the central axis direction should be inside the square end. Otherwise, the wind loss at the air outlet should be relatively large. Therefore, the maximum diameter of the round end was φ 180 mm, and the analysis was shown in Figures 7 and 8,

(1)    Import $2.15 \times 10^2 \times 1.8 \times 10^2$ mm, export φ $1.8 \times 10^2$ mm
(2)    Import $2.15 \times 10^2 \times 1.8 \times 10^2$ mm, export φ $1.6 \times 10^2$ mm

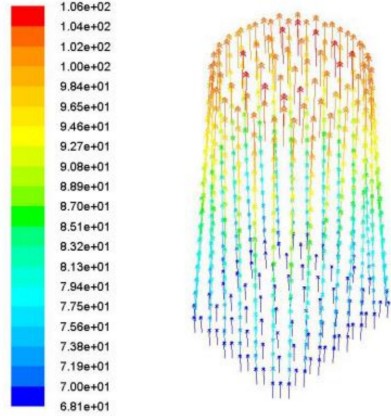

**Figure 7.** Simulation of φ $1.8 \times 10^2$ mm airduct.

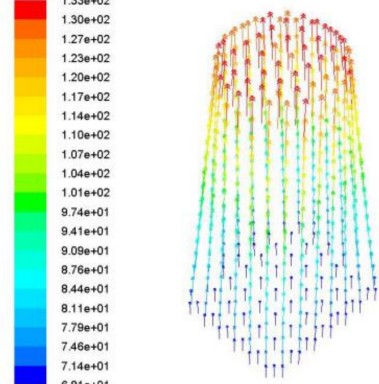

**Figure 8.** Simulation of φ $1.6 \times 10^2$ mm airduct.

*2.6. Dynamic Simulation Analysis of Soil Particle Group Under Wind Blowing*

2.6.1. Soil Ridge Modeling and Flow Field Modeling

The soil particle model was added to the particle factory of EDEM [21], according to the field measurement of the soil ridge size in the core area of the wine grapevine, as shown in Figure 9a, then a 2 m long soil-covered ridge model was established in EDEM, and the cross-section was an isosceles trapezoid with an upper base of 350 mm, a lower base of 600 mm and a height of 300 mm, as shown in Figure 9b.

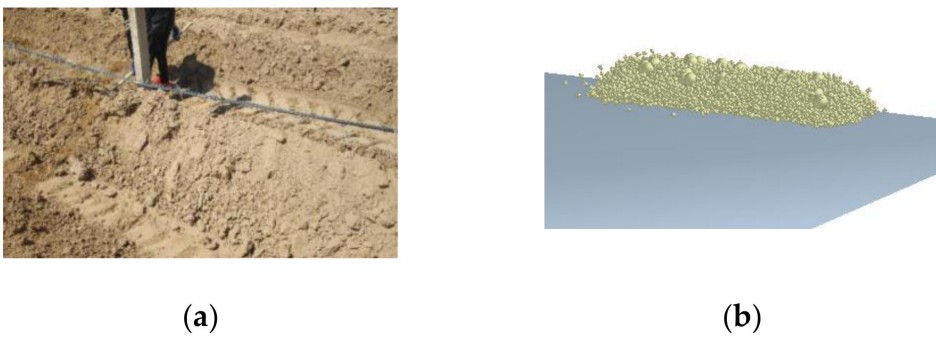

(**a**) (**b**)

**Figure 9.** (**a**) Field model. (**b**) Simulation model.

Because the simulated soil ridge was the shape of the soil ridge in the core area of the rattan edge after mechanical cleaning, it was an unnatural shape. However, it cannot be accumulated into this shape by virtue of free-falling soil particles in EDEM software. Therefore, a three-dimensional model of a trapezoidal groove was first imported in Geometries, as shown in Figure 10. The trapezoidal groove was set as a solid, and a particle factory was established at the upper opening to make the particles fall freely and accumulate in the groove in a corresponding shape. After stacking was completed, the entity slot was set to virtual or delete, and continued the subsequent simulation.

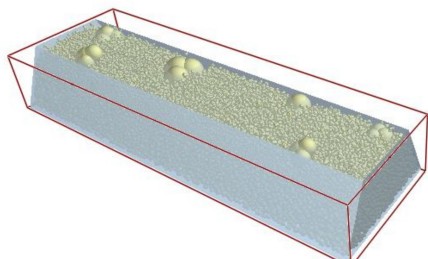

**Figure 10.** Trapezoid groove.

Based on the field conditions and experimental basis of vineyards in Ningxia province of China, a geometric model of the flow field space was established, the flow field model was divided into grids and the air inlet, wall and air outlet were set up. According to the scheme demonstration test, the maximum fall point range of the soil was about 5 m from the air outlet, so the main size of the flow field space was set to a rectangular parallel pipe with a base length of 5 m and a height of 3 m, as shown in Figure 11, and the size of the air inlet of the flow field was the size of the air outlet of the airduct.

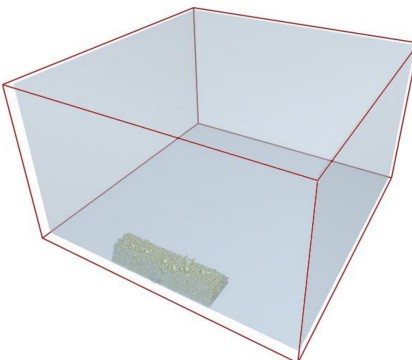

**Figure 11.** Geometric model of the flow field.

2.6.2. Simulation Process

The Fluent-EDEM coupling simulation of soil under the action of wind blowing [22–25] was used to study the airduct opening at a certain pointing angle. The airduct opening center was aligned at 1/2 of the height of the soil ridge. The straight-line distance between the airduct mouth and soil ridge was 100 mm. The rate of damage to the soil-covered ridge under the action of wind was obtained. The soil clearance rate within the effective working time was obtained as ten. Assuming that the mass of the original soil ridge was $m_0$, and the mass of the soil ridge after cleaning the soil was $m_1$, the calculation formula of the soil clearance rate, c, was as shown in Formula (17):

$$c = \frac{m_1 - m_0}{m_0} \qquad (17)$$

It took 5 s to generate the soil ridges; from this point on, the coupling was carried out, wind was added, and the simulation proceeded to the 7th second. By using EDEM software simulation post-processing, the effect of soil cleaning can be seen intuitively through the change trend of the soil ridge quality and shape. The mass of the soil ridge in the red area in front of the airduct opening was 147.572 kg, as shown in Figure 12. Three positions of the duct opening orientation angles, 0°, 30° and 45°, were selected for research. The simulation and calculation showed that the wind speed of the duct was 68 m·s$^{-1}$.

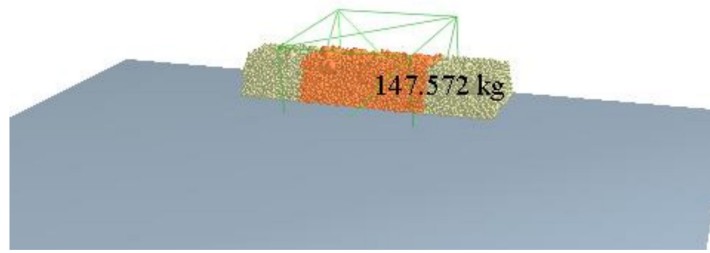

**Figure 12.** Initial soil ridge quality.

The purpose of this simulation was to find the best wind-bearing point of the scattered soil ridge by changing the angle of the air outlet under the condition that the center position of the air outlet remains unchanged. Since the height of the airduct was adjustable, the height of the air outlet did not change when the angle of the air outlet changed. As shown in Figure 13, the air outlet angle included three components: 0° (in blue), 30° (in pink) and 45° (in green).

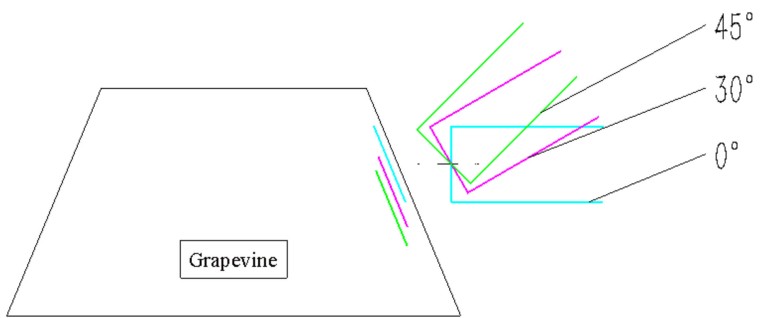

**Figure 13.** Schematic diagram of air outlet angle.

When the direction angle of the airduct was 0°, 30° or 45°, the change of soil ridge quality at different times was as shown in Table 5, and the simulation process is shown in Figures 14–16.

**Table 5.** The change of soil ridge quality with different direction angles of the duct mouth.

| The Direction Angle of the Duct Mouth | Initial Mass of Soil Ridge/(kg) | Soil Ridge Mass at 0.5 s/(kg) | Soil Ridge Mass at 1.5 s/(kg) | Soil Ridge Mass at 2 s/(kg) |
| --- | --- | --- | --- | --- |
| 0° | 147.572 | 142.248 | 116.19 | 115.008 |
| 30° | 147.572 | 137.782 | 97.977 | 94.5624 |
| 45° | 147.572 | 144.419 | 113.904 | 110.696 |

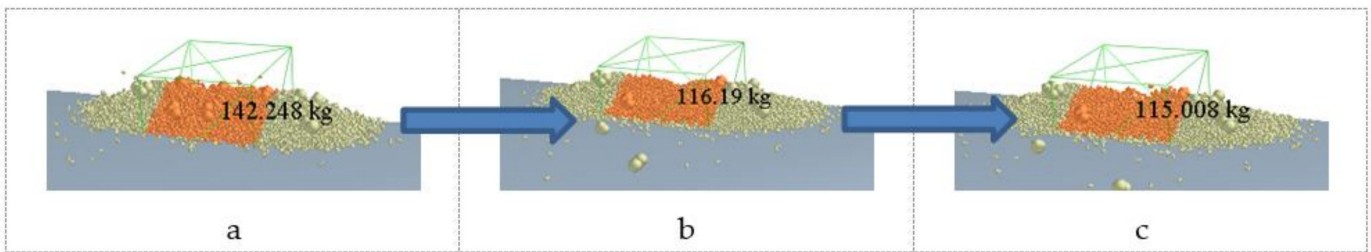

**Figure 14.** Simulation process of airduct mouth direction angle, 0°. (**a**) soil ridge mass at 0.5 s/(kg), (**b**) soil ridge mass at 1.5 s/(kg), (**c**) soil ridge mass at 2 s/(kg).

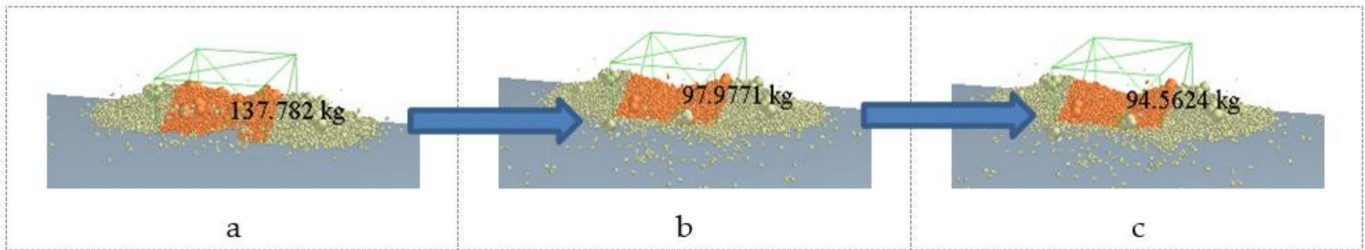

**Figure 15.** Simulation process of airduct mouth direction angle, 30°. (**a**) soil ridge mass at 0.5 s/(kg), (**b**) soil ridge mass at 1.5 s/(kg), (**c**) soil ridge mass at 2 s/(kg).

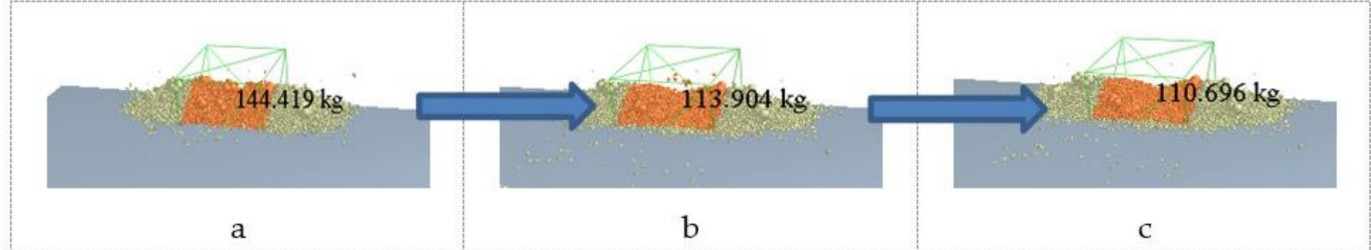

**Figure 16.** Simulation process of airduct mouth direction angle, 45°. (**a**) soil ridge mass at 0.5 s/(kg), (**b**) soil ridge mass at 1.5 s/(kg), (**c**) soil ridge mass at 2 s/(kg).

The simulation process of ground removal with grapevines inside the soil ridge is reported hereinafter. The trunk of the vine was 3 m long and 3 cm in diameter, with branches on the side, the vines in the picture were pink, and the simulation was the same as the above the process, and is shown in Figures 17–19. The change of soil ridge quality at different times is shown in Table 6.

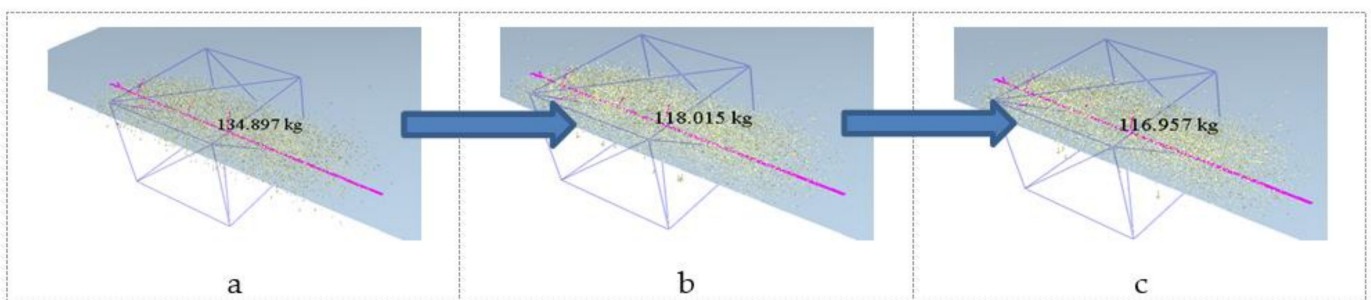

**Figure 17.** Simulation process of airduct mouth direction angle, 0°. (**a**) soil ridge mass at 0.5 s/(kg), (**b**) soil ridge mass at 1.5 s/(kg), (**c**) soil ridge mass at 2 s/(kg).

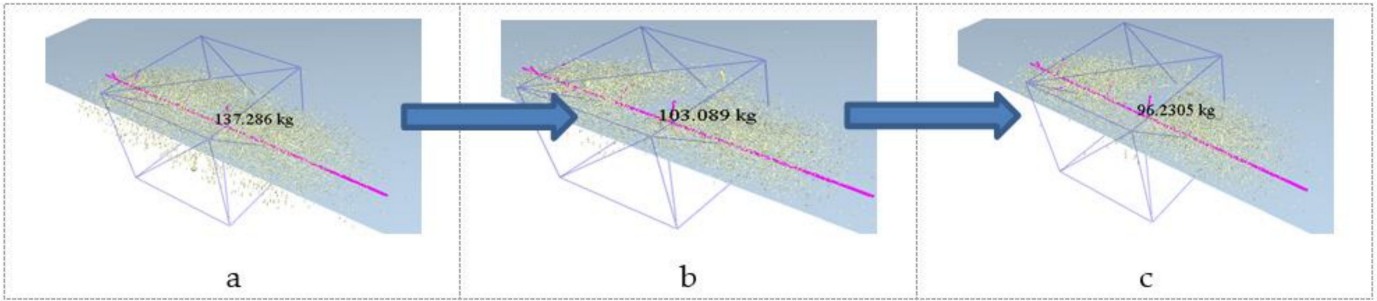

**Figure 18.** Simulation process of airduct mouth direction angle, 30°. (**a**) soil ridge mass at 0.5 s/(kg), (**b**) soil ridge mass at 1.5 s/(kg), (**c**) soil ridge mass at 2 s/(kg).

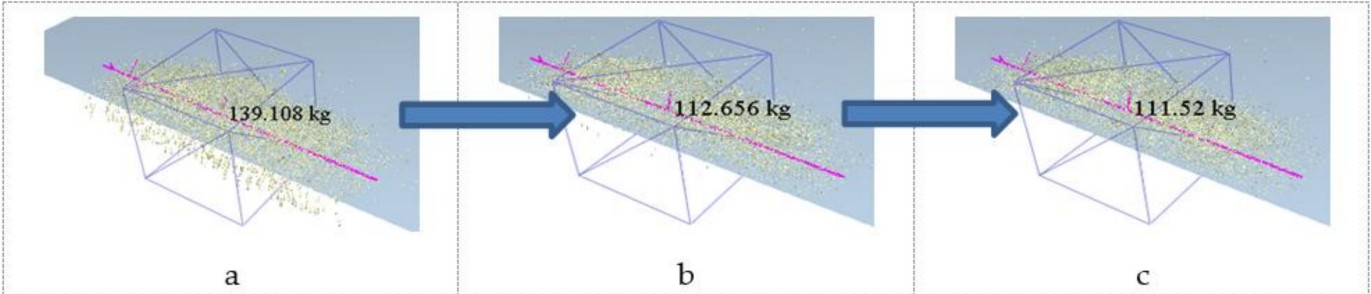

**Figure 19.** Simulation process of airduct mouth direction angle, 45°. (**a**) soil ridge mass at 0.5 s/(kg), (**b**) soil ridge mass at 1.5 s/(kg), (**c**) soil ridge mass at 2 s/(kg).

**Table 6.** The change of soil ridge quality with different direction angle of the duct mouth.

| The Direction Angle of the Duct Mouth | Initial Mass of Soil Ridge/(kg) | Soil Ridge Mass at 0.5 s/(kg) | Soil Ridge Mass at 1.5 s/(kg) | Soil Ridge Mass at 2 s/(kg) |
|---|---|---|---|---|
| 0° | 147.572 | 134.897 | 118.015 | 116.957 |
| 30° | 147.572 | 137.286 | 103.089 | 96.2305 |
| 45° | 147.572 | 139.108 | 112.656 | 111.520 |

Based on the above results, it can be seen that the simulation results of soil ridges with vines and soil ridges without vines were roughly the same, and the vines had little effect on the air-blown wind cleaning.

### 2.7. Development and Test of Principle Prototype

2.7.1. Development of the Principle Prototype

According to theoretical calculations, the principle prototype is as shown in Figure 20. It used a 22 kW high-pressure centrifugal fan. The whole machine consisted of a centrifugal fan, chassis, windpipe, hose, fan outlet transfer tube and windpipe adjustment device. The chassis installation platform was 600 mm from the ground, with a three-point suspension mounting frame in the front, and a pair of depth-limiting wheels at the rear to support the whole machine during placing or dragging operation. The centrifugal fan was fixed on the chassis, and the fan outlet transfer tube, that is, the combined pipe, was fixed to the fan air outlet with bolts. The windpipe was a metal hard pipe, and the hose was made of PVC material. PVC plastic, chemical field referred to compound polyvinyl chloride. The hose connected the air outlet of the fan with the air inlet end of the windpipe. The windpipe adjustment device was fixed on the side of the chassis, holding up the windpipe, and adjusted the length of the hose and the direction of the windpipe by adjusting the position of the bolt on the lead screw. Due to the large power of the centrifugal fan, the tractor cannot supply power for it. Therefore, an 80 kW generator was used as a power source with a 30 m long wire to provide a 380 V three-phase alternating current for the centrifugal fan motor. The wind duct was adjusted to a 30° angle, and the prototype was towed by the tractor.

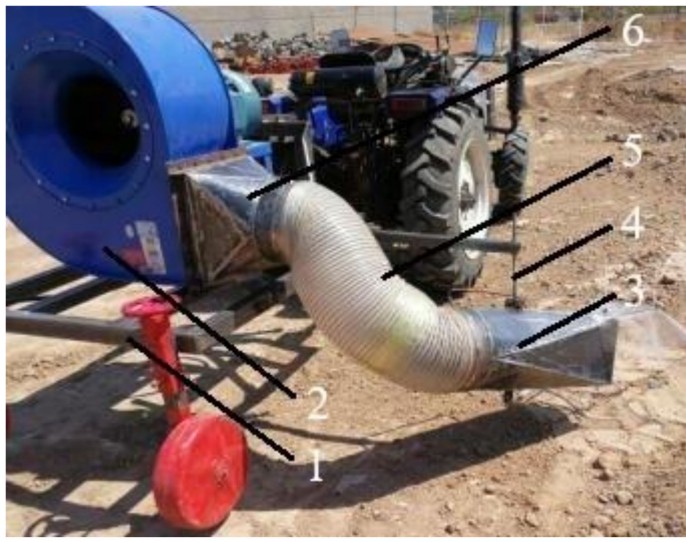

**Figure 20.** Principle prototype. 1: chassis, 2: centrifugal fan, 3: windpipe, 4: fan outlet transfer tube, 5: hose, 6: windpipe adjustment device.

### 2.7.2. Principle Prototype Test Method

In order to test the soil-cleaning effect of the selected 22 kW high-pressure centrifugal fan and whether the principle prototype would damage the vines, the prototype was tested in Ningxia, China, based on theoretical analysis and simulation results. A field simulation test was carried out with a 90-horsepower tractor pulling the principle prototype, and then a field performance test was carried out in a wine-making vineyard in Ningxia, China.

### 2.7.3. Field Simulation Test

In the first group of experiments, 5 m long soil ridges were piled up and compacted according to the size of the soil ridges of the vineyard given by the project in Ningxia region of China to test the cleaning effect. In the second group of experiments, the soil in the nearby fields (similar to the vineyard soil) was used to pile up soil ridges according to the size of vine edge core area measured in the field of grape plantation in Ningxia, China. During stacking, the soil was watered layer by layer, so that the moisture content inside the soil ridges was similar to the field. After the stacking was completed, there was a 3 h wait to make the surface of the soil ridge was dry and the overall strength was not lower than that of the core area of the vine side in the grape plantation.

The prototype was fixed on the rear of the tractor by suspension, the 380 V power supply of the workshop was connected with the centrifugal fan motor with wires and the tractor was driven to the working position beside the soil ridge. The power supply was turned on until the speed of the fan was stable and the tractor started to move at a constant speed of about 0.5 m·s$^{-1}$, and then the soil cleaning effect and the damage of the vines was observed.

### 2.7.4. Field Test on Grape Plantation of Yuquanying Farm in Ningxia, China

The field test was selected on a grape plantation in China's Ningxia region where the mechanical (scraper) cleaning was completed just one day ago. After the tractor pulled the prototype to the working position, the direction and position of the airduct were adjusted, the fan was started and there was a wait for the speed to stabilize, then the tractor started to move forward at a speed of about 0.5 m·s$^{-1}$ and the effect of soil cleaning was observed and the range of soil dispersion was measured.

Table 7 shows the significance of symbols in this paper.

**Table 7.** Nomenclature.

| Symbol | Significance | Unit |
|:---:|:---:|:---:|
| φ | The static friction angle | ° |
| nu | The soil Poisson's ratio | none |
| $K_0$ | The soil lateral pressure coefficient | none |
| Ø | The effective static friction angle of soil | ° |
| G | The soil shear modulus | $MN \cdot m^{-2}$ |
| E | The soil elastic modulus | kPa |
| v | The moving speed | $m \cdot s^{-1}$ |
| V | The volume | $m^3$ |
| m | The mass | kg |
| $\rho_1$ | The soil density | $kg \cdot m^{-3}$ |
| $F_1$ | The force required to push the soil ridge | N |
| s | The covering soil ridge was pushed horizontally along the width | m |
| P | The minimum power required to push the soil ridge | kW |
| $W_1$ | The work required to achieve the above effect | J |
| $W_2$ | Kinetic energy of the fan | J |
| $\rho_2$ | Air density | $kg \cdot m^{-3}$ |
| Q | Air volume | $m^3 \cdot h^{-1}$ |
| $v_0$ | Air outlet speed of fan duct | $m \cdot s^{-1}$ |
| S | Cross-sectional area of air outlet | $m^2$ |
| I | The wire current | A |
| P′ | Motor-rated power | kW |
| U | Voltage | V |

## 3. Results

### 3.1. Simulation Analysis of the Square Tube

In the optional range of this scheme, when the air inlet and outlet were both 215 × 180 rectangular tubes, the wind speed of the air outlet only slightly increased to 68.9 $m \cdot s^{-1}$ in the middle area, and the wind speed decreased in most areas around. The wind speed loss was large, especially in the edges. When the inlet was 215 × 180 mm and the outlet was 180 × 180 mm tapered square tube, the wind speed at the outlet could reach 80 $m \cdot s^{-1}$. When the inlet was 215 × 180 mm and the outlet was 160 × 160 mm tapered square tube, the wind speed was further increased, and the wind speed at the outlet could reach 103 $m \cdot s^{-1}$, which is shown in Table 8.

**Table 8.** Wind speed value.

| Duct Size | Wind Speed Value ($m \cdot s^{-1}$) |
|:---|:---:|
| Both air inlet and outlet were 215 × 180 mm rectangular tubes | 68.9 |
| Inlet 215 × 180 mm, outlet 180 × 180 mm tapered square tube | 80 |
| Imported 215 × 180 mm, outlet 160 × 160 mm tapered square tube | 103 |

### 3.2. Simulation Analysis of the Combined Taper Pipe

In the optional range of this scheme, the combined taper pipe had a greater effect on the wind speed than the square pipe. When the outlet was φ 180 mm, the theoretical outlet wind speed could reach 106 $m \cdot s^{-1}$. When the outlet was φ 160 mm, the theoretical outlet wind speed at the outlet could reach 133 $m \cdot s^{-1}$.

### 3.3. Simulation Results of Duct Fluid

The simulation results showed that the smaller the size of the outlet pipe, the greater the wind speed of the outlet pipe and the greater the dynamic pressure obtained on the target external surface. Since the wind speed obtained by the simulation was the theoretical wind speed, in fact, the friction between the air and the inner wall of the pipe would cause

energy loss, and the final outlet wind speed would be lower than the theoretical value. In order to ensure the efficiency of soil cleaning, a combined-type pipe with air outlet size of φ 180 mm was selected.

### 3.4. Result Analysis of Soil Particle Group under Wind Blowing

Since the simulation results of soil ridges with vines and soil ridges without vines are roughly the same under the action of wind blowing, the subsequent results were analyzed with soil ridges without vines.

By using EDEM software to draw a line graph of "mass-time" and observe the change of soil ridge quality with the time of wind action, the soil clearance rate can be calculated conveniently. Since the horizontal dimension of the airduct opening was determined to be 180 mm and the operating speed was 0.42 m·s$^{-1}$, the effective time of the wind acting on the front of the duct opening was about 0.43 s, that is, 0.43 s was taken as a period for the study.

As shown in Figure 21, the mass of the original soil ridge was 292.3 kg. Figure 21a shows the soil-cleaning situation when the windpipe mouth was pointed at 0°. At 0.43 s, the mass of the remaining soil was about 285 kg, that is, the single soil-cleaning rate was 2.5%; at 0.86 s, the mass of the remaining soil was about 251 kg, that is, the round-trip soil-cleaning rate was 14.1%; at 1.72 s, the mass of the remaining soil was about 213 kg, that is, the soil-cleaning rate between the two rounds was 27.1%.

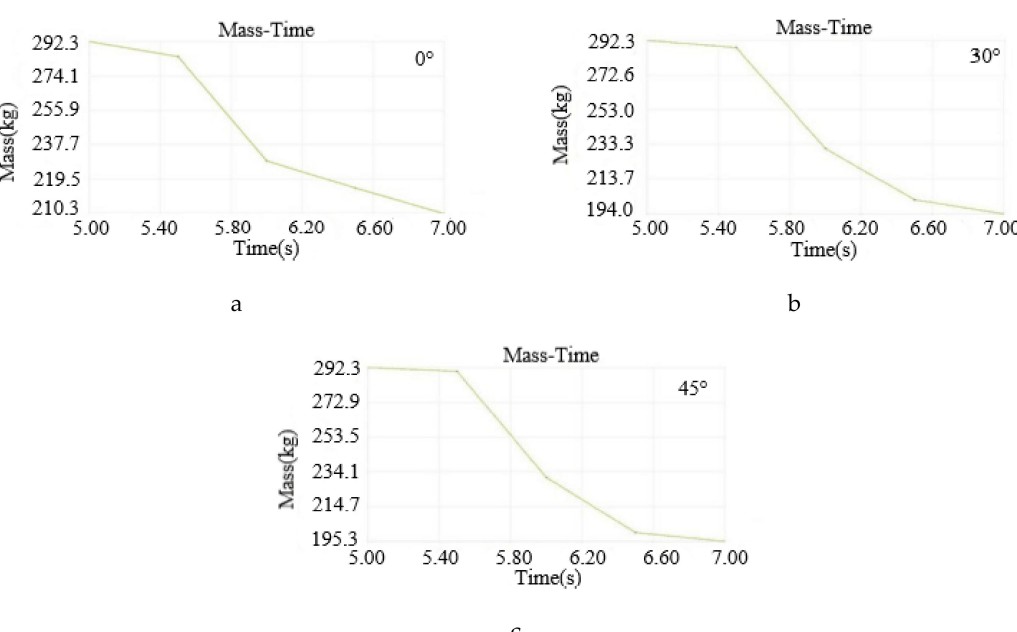

**Figure 21.** Mass-Time line chart. (**a**) the soil-cleaning situation when the windpipe mouth was pointed at 0°, (**b**) the soil-cleaning situation when the windpipe mouth was pointing at an angle of 30°, (**c**) the soil-cleaning situation when the windpipe mouth was pointing at an angle of 45°.

Figure 21b shows the soil-cleaning situation when the windpipe mouth was pointing at an angle of 30°. At 0.43 s, the mass of the remaining soil was about 288 kg, that is, the single soil-cleaning rate was 1.5%; at 0.86 s, the mass of the remaining soil was about 252 kg, that is, the round-trip soil-cleaning rate was 13.8%; at 1.72 s, the mass of the remaining soil was about 200 kg, that is, the soil-cleaning rate between the two rounds was 31.6%.

Figure 21c shows the soil-cleaning situation when the windpipe mouth was pointing at an angle of 45°. At 0.43 s, the mass of the remaining soil was about 289 kg, that is, the single soil-cleaning rate was 1.1%; at 0.86 s, the mass of the remaining soil was about 255 kg,

that is, the round-trip soil-cleaning rate was 12.8%; at 1.72 s, the mass of the remaining soil was about 201 kg, that is, the soil-cleaning rate between the two rounds was 31.2%.

Figure 22 shows the soil-cleaning situation when the direction angles of the airduct were 0°, 30° and 45°. The change of soil ridge quality at different times is shown in Table 9.

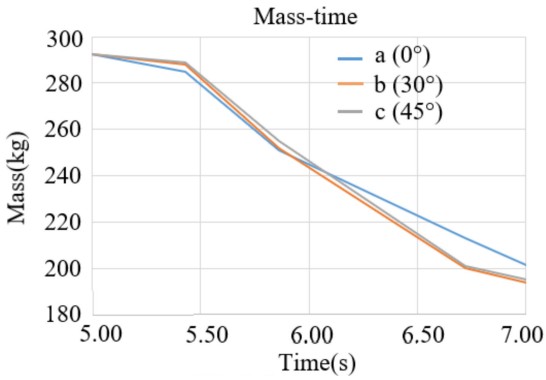

**Figure 22.** Soil-cleaning situation.

**Table 9.** The change of soil ridge quality with different direction angles of the duct mouth.

| The Direction Angle of the Duct Mouth | Initial Mass of Soil Ridge/(kg) | Soil Ridge Mass at 0.43 s/(kg) | Soil Ridge Mass at 0.86 s/(kg) | Soil Ridge Mass at 1.72 s/(kg) | Soil Ridge Mass at 2 s/(kg) |
|---|---|---|---|---|---|
| 0° | 292.3 | 285.0 | 251.0 | 213.0 | 201.3 |
| 30° | 292.3 | 288.0 | 252.0 | 200.0 | 194.0 |
| 45° | 292.3 | 289.0 | 255.0 | 201.0 | 195.3 |

According to the line chart, the soil quantity decreased slowly within 0.5 s after the start of wind cleaning, and only a small amount of soil particles were blown away. From 0.5 to 1.5 s, the soil quantity decreased rapidly, and the ridge collapsed rapidly under the action of wind blowing. After 1.5 s, the rate of soil mass reduction slowed down again, and the soil under the ridge was not easy to be dispersed due to the size and location of the air outlet. When the direction angle of the airduct was 0°, the soil removal rates of a single operation and a round-trip operation were the highest, which were respectively 2.5% and 14.1%. When the direction angle of the airduct was 30°, the two round-trip soil removal rates were the highest, which was 31.6%.

Although the soil removal rate was the highest in a single operation and one round-trip operation when the airduct opening angle was 0°, in the actual grape plantation operations, the airduct opening angle was too small, which would make it difficult to install and arrange the fan and airduct, and the airduct was close to the ground during the operation, which can easily cause damage to the airduct. Therefore, the direction angle of the duct opening should be appropriately increased in practical applications. When the pointing angle was 30°, the soil clearance rates of a single operation and a round-trip operation were not much different from the 0° pointing angle. The actual prototype would adopt a 30° windpipe orifice pointing angle.

### 3.5. The Results of Tests

3.5.1. The Results of Field Simulation Test

① Under the action of wind, the surface layer of floating soil was immediately blown away, and the overburden in the central area of the vine edge was subsequently stripped and collapsed layer by layer, which was consistent with the simulation results.

② In the first group of site simulation tests, the soil ridge with a low water content had a good soil-cleaning effect, with a soil-cleaning rate of over 70%.

③ In the second group of field simulation tests, when the whole machine moved at a speed of 0.5 m/s, the average soil removal rate in the core area of the vines could reach 80%,

that is, the wind force of the fan can ensure that most of the covering soil above the vines was blown away, exposing the vines, and a few areas with loose soil can even be completely cleaned.

### 3.5.2. The Results of Field Test on Grape Plantation of Yuquanying Farm in Ningxia, China

The soil ridge before the test was the soil left after scraping once with the scraper, which was the core area soil ridge. The cross-sectional area of the soil ridge was trapezoidal. After the scraper was cleaned up, the remaining core area was due to the collapse of the soil. The cross-sectional area of the soil ridge was not very uniform. Five measurement points were taken at random, the position of each point was recorded with colorful flags, the upper width, lower width and height of each soil ridge were measured before and after the soil was cleaned and the cross-sectional area of the soil ridge was calculated. In the field test of the grape plantation in Ningxia, China, due to the large bending and small row spacing of the hose part of the airduct during processing and manufacturing, the hose could not be fully deployed, and the generator power was unstable, resulting in poor soil-cleaning effect. When the soil-cleaning speed was 0.5 m·s$^{-1}$, the soil-cleaning rate was only about 15%, and the vines could not be exposed, as shown in Figure 23b. When the hard tube of the tuyere was removed and the hose was straightened to the soil ridge, the cleaning effect was greatly improved. The sizes of the soil ridges at the marked measuring points were measured with a tape measure, and the cross-sectional area was calculated before and after the soil cleaning. The experimental results are shown in Table 10, and the schematic diagram is shown in Figure 23c,d.

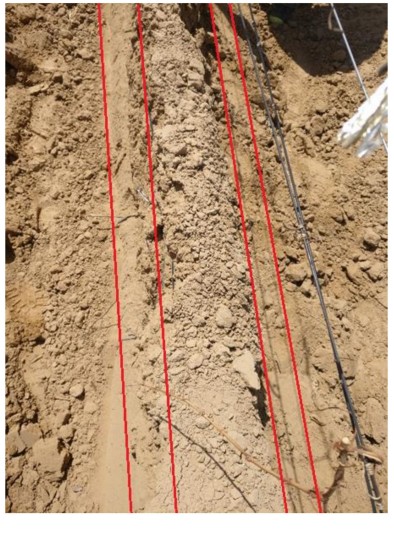

a

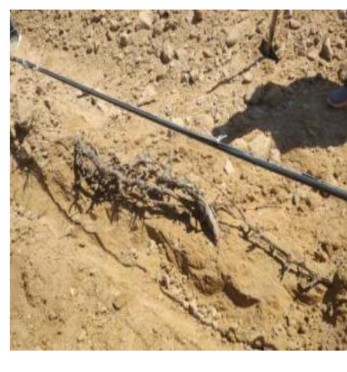
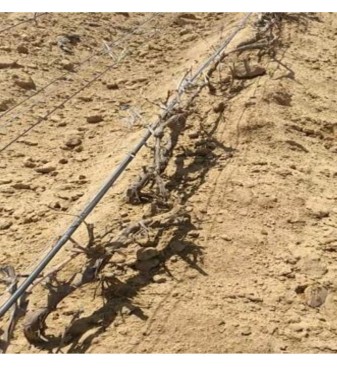
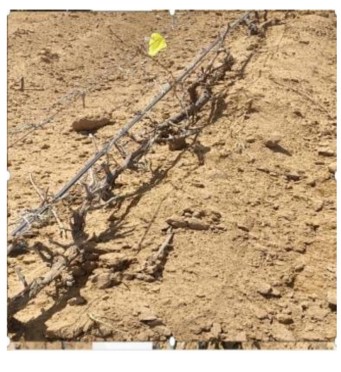

b c d

**Figure 23.** Soil cleaning effect. (**a**) the soil ridge when the scraper scrapes over, (**b**) experimental results without removing the hard tube, (**c**,**d**) experimental results with removing the hard tube.

**Table 10.** Results of performance experiment.

| Wind-Blowing Type Soil Cleaner | | |
|---|---|---|
| Cross-Sectional Area before Cleaning/cm$^2$ | Cross-Sectional Area After Cleaning/cm$^2$ | Cleaning Efficiency/% |
| 1111 | 245 | 77.95 |
| 1502 | 354 | 76.43 |
| 1425 | 328 | 76.98 |
| 1175 | 260 | 77.87 |
| 1645 | 390 | 76.29 |
| Average cleaning efficiency/% | | 77.10 |

### 3.5.3. Analysis of Test Results

Both tests were able to successfully blow off the soil covering the core area around the wine grapevines, which proved the reliability of the scheme. In the field test of the grape plantation in Ningxia, China, it was found that the airduct opening should be as close as possible to the soil ridge, which can reduce the air flow at the airduct opening and increase the soil removal rate. After the scraper was scraped, the soil would obviously loosen and collapse, so the field cleaning efficiency can reach about 77%, the exposed vine rate can reach more than 80% and the exposed vines were completely exposed to the outside. The simulation test investigated the quality of the overall soil ridge. The soil removal rate can reach more than 30%, which proved that the air-blowing scheme can remove about 30% of the soil. The current scraper type cleaning machine can scrape about 60% of the soil, so using this scheme to blow away the core area of the soil is feasible, and finally, can fully reveal the grapevine.

### 4. Discussion and Conclusions

In this paper, a wind-blown wine grape cleaning method was proposed for the major grape-growing countries in Central Asia that require vine burial in winter and vine planting in spring, and a prototype machine was developed to remove the soil covering the core area of the wine grapevines. A prototype test was conducted in northwestern China, and the machine was small in size, light in weight and had a significant cleaning effect. The main conclusions were as follows:

(1) Combined with the climate, soil and wine grape planting patterns in Central Asia, the overall design requirements were proposed. The designed principle prototype included a blower, a power device and a chassis. Based on the overall design requirements and through a field demonstration, the overall design of the wind-blown wine grape cleaning machine suitable for Central Asia was determined.

(2) Through the experiments, the physical and physical property parameters of the wine-growing vineyard soil in Ningxia, Northwest China, were measured, and the soil particle model and the soil-covered ridge model were established in the EDEM software. Through the Fluent-EDEM coupling method, the dynamic simulation analysis of the soil of the wine grape plantation under the action of wind-blowing was completed, and the original design parameters of the wind-blowing device were obtained.

(3) Based on the overall design scheme and the size of the soil ridges in the core area around the vines of the wine grape plantations in Northwest China, the prototype was developed and tested. The average cleaning speed was 0.5 m·s$^{-1}$. The average soil-cleaning rate was stable at about 80%, which could completely reveal the grapevine without hurting the vines or buds.

This method greatly reduced the labor cost of vines for wine grapes in spring, greatly improved the efficiency of soil ridge vines in the core area and provided the possibility of mechanized vines for wine grapes in Central Asia in spring.

**Author Contributions:** Conceptualization, Methodology, Resources, Supervision, Q.Y.; Software, Writing—Review and Editing, M.H.; Formal analysis, Investigation, G.D., Validation, L.S.; Writing—Original Draft, A.S.; Data Curation, X.Z.; Visualization, M.A. All authors have read and agreed to the published version of the manuscript.

**Funding:** This work is supported by the National Natural Science Foundation of China (51675239); The National Science and Technology Major Project of China (2019YFD1002502); The Natural Science Fund Project of Colleges in Jiangsu Province of China (19KJA430018); The Important Development Program of Ningxia Province of China (2018BBF02020); The Research and Development Program of Zhenjiang Province of China (NY2019015).

**Institutional Review Board Statement:** This article did not involve human or animal research.

**Informed Consent Statement:** This article did not involve human or animal research.

**Data Availability Statement:** Not applicable.

**Acknowledgments:** The authors would like to thank their schools and colleges, as well as the funding of the project.

**Conflicts of Interest:** We declare that we have no conflict of interest.

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
