# Peer review of "Design and Experimental Study of a Wine Grape Covering Soil-Cleaning Machine with Wind Blowing"

_agriengineering, doi:10.3390/agriengineering3010004_

Round 1

Reviewer 1 Report

1. section 2.2.1
- I suggest to improve the description of attachment to the tractor. In 2.5.1 there is a more detailed description explaining that it is suspended from the 3-points and the wheels are to limit height.
2. section 2.2.2.
- Could the power take-off be used to move the fan using a gearbox or pulleys to fit the fan rpm's?
3. section 2.3.2.1
- In my humble opinion, this section should be improved:
* a) I recommend to use "airflow" or "air flow rate" inteads of "air volume" when referring to volumen of air per unit of time (m3/h) and "mass flow" or "mass flow rate" when referring to mass per unit of time. (This would be applicable to all the sections).
* b) Although in the text it's explained that this calculations apply to 1-second time, I think that formulation could be improved.
* c) I propose to provide explicitly the value of the required power in kW instead of the Energy for 1-second time.
* d) I understand that the values for volumen calculation correspond to a triangle section of 0.2x0.4, however in 2.4.1 the ridge section is modeled like a trapezoide (350/600 x 300 mm high). Why are they different?
* e) In formula 2.10 and 2.11 Q is the airflow rate and F the outlet area. F dimensions are not clearly defined although from the line number 196 I guess they are 0.2 and 0.42, corresponding to S. I guess that 0.42 m come from the 1-second time, as the distance swept by the fan in 1-second at 0.42 m/s speed.
* f) I think that S like "Area per unit of time" should be F.
* g) Equations "in-line" should be avoided (e.g. line number 186,195)
* h) air density shows a Chinese symbol. Air density value should be indicated.
4. line number 210:
- wrong reference
1. section 2.5.3 and 2.5.4
- Redaction should be improved, some paragraphs look as copy-paste from a procedure (lines 305, 308-311 and 316-318)
2. Section 3.1
- Redaction could be simplified. I recommend a table with air speed values for each size and to summarize the results in one paragraph, as it has been don in section 3.2
3. section 3.4 lines 70-371
- a table would be useful
4. section 3.5.3
- lines 424-428 are a repetition of section 3.5.2. I suggest to summarize it.
5. Section 4
- I propose to simplify lines 436 to 439. In my opinion expression "designed wind blowing type wine grape cleaning principle prototype" could be simplify.

Reviewer 2 Report

GENERAL COMMENT

The article deals with a study on a wine grape covering soil cleaning machine and in particular it illustrates a sort of development process that lead to the final design for this machine. The topic could be interested but many issues need to be solved and some corrections/integrations should be performed following the comments reported here, before this article comes to a form suitable to be published.

Firstly, there are some problems in the calculations (see specific comments) and in how the moisture content has been considered.

Then, concerning the design, there is a thing that is not clear: why exposing the fluid dynamic simulations with Fluent on the air duct before the DEM simulations? I think that a correct way of operating is using the fields of motion of the air (from the Fluent simulations) as a starting point to simulate the particle removal on the soil ridge. I also suggest the authors to include in their simulations also the vine plants (maybe with a standardized shape), as they has a sure influence in retaining the soil.

Finally, the way the field experimentation was led rises many problematic points: as it was reported in the article, the experimentation seems to be almost useless, because many parameters were out of the control of the experimenters and, also, no metric was set to quantify the progress of the soil removal from the ridge during the operation of the machine. My suggestion is to repeat the experimentation and fill the important gaps highlighted above. About the progressive quantification of the machine effect, I suggest, for example, to use a visual support (e.g. photographs) to measure the ridge profile in set time intervals and quantify ex post the volume (hence the mass) of removed soil. The best way would be put all the terrain (and the vine plants) on a scale and measure with continuity the mass decrement over the time.

SPECIFIC COMMENTS

  • Introduction: as you use many symbols in your article, please add a nomenclature table reporting all the symbols, together with their measurement units and a short explanation of the referred physical quantity.
  • Introduction: please could you add a photograph of the grape burial operation taken during the last season? As this operation is not so common, it could be interesting also include in the article a picture (about near line 42).
  • Please indicate the measurement units in the form of power (e.g., m2/h -> m2 h-1) and put always a space between the value and the measurement units (in general, please follow the NIST guidelines to properly handle the SI: https://physics.nist.gov/cuu/pdf/sp811.pdf)
  • Pay attention that contracted forms (e.g. “isn’t”, “couldn’t”) are not used in formal English (i.e. the English that should be adopted for an article) and, therefore, should be avoided.
  • Table 2.1: instead of “project name” as header I suggest to use “Technical parameter of the machine”
  • Table 2.1: please could you add some parameters more? E.g.: the electrical parameters of the motor (power, voltage, number of poles, speed range…), the fan impeller diameter, the air duct section at the end of the airduct, the air speed range. Specifically, I would add to this table also the parameters that were object of study/optimization, clearly underlining this and indicating also the investigated values or range of values (in the simulations or in the field experimentation).
  • Table 2.1: I think that “operating speed” is referred to the advancement speed of the machine beside a ridge. Please refer to it rather as “machine advancement speed” to avoid any confusion with the (mean) air speed.
  • Lines 98-100: please add also the colour of each subsystem you are referring to in the text, for an easier reference; please do the same in the figure caption.
  • Line 143: “Take Ningxia, one of the main wine grape producing areas in Northwest China, as an example. The soil moisture content measured by oven drying at temperature of 105°C for 72 hr.” -> “In this paragraph, Ningxia, one of the main wine grape producing areas in Northwest China, has been taken as an example. The soil moisture content was measured by drying samples in an oven at temperature of 105 °C for 72 h.” At this regard, it could be appropriate to cite also the international norm(s) followed for the procedure of soil moisture determination; the same should also be done for the ring-knife method, the elastic modulus calculation, the static friction angle (and all other experimental procedures).
  • Table 2.2: please use the same number of significant figures for all the presented numbers (pay attention to the last column in particular).
  • Lines 153-154: according to my experience, the Poisson's ratio is usually indicated with the Greek letter “nu” (ν); please check.
  • Table 2.4: pay attention that, for the table layout chosen by you, the label “Particle size name” is referred to the items of the same column, if you want to refer to the quantities in the same row (e.g., “Clod - 10mm aggregate - 5 mm aggregate - Fine sand”) you have to put that label in a row above these ones.
  • Line 171: please make explicit the calculations which led you to obtain the quantity 4669 m2/h. Also, it is not correct call it “operating speed” because it has not the physical dimensions of a speed; please use instead another name (e.g. area of the vineyards to be processed in a reference time period, in this case an hour).
  • Equations 2.5 and following ones (please check also throughout the article): please put the measurement units also beside the quantities reported in the intermediate passages of your calculations, not only in the final results.
  • 2.9: there is a thing that is not clear at all, how did you calculate the work to be done if the weight force acts along a vertical direction and the displacement is along a perpendicular direction? As the angle between these two lines of action is 90°, the work should be null. I think you have to consider the friction coefficient multiplied by the gravity force. Please check and correct the calculations. Also, how did you consider the moisture level in determining the fore to be overcome by the air stream? In my opinion it should have a great influence (causing adhesion amongst soil particles).
  • 2.10 and 2.11: pay attention, in my pdf there is a strange symbol (Chinese?) as subscript after the symbol of the density.
  • Line 192: “pa” -> “Pa”
  • Lines 200-204: please specify in the text also that you have used a 1.5 over-dimensioning coefficient for the sizing of the electric wires.
  • Line 264: a curiosity, when you changed the angle of the air stream, did you modify also the height of the air outlet to proper collide with the ridge of soil? If yes/no please specify. Could you also add a scheme?
  • Line 293: “15KW” -> “15 kW” (“K” is a small letter)
  • Paragraph 3.1. Please evidence that your simulations have a fundamental assumption, that is a complete absence of influence of the fan impeller on the air stream (in terms of: no turbulence, no flow deviation from the direction parallel to the airduct axis…)
  • Line 359: “and the greater the dynamic pressure.” -> “and the greater the dynamic pressure obtained on the target external surface, i.e. on the soil ridge” (correct as indicated just to avoid any possible misunderstanding with the static pressure on the pipe walls)
  • 3.6: please present a single graph with three curves (and a legend about the angles), in this way comparisons are immediate also at the readers’ eyes.

Round 2

Reviewer 2 Report

  • Nomenclature table: please add also the measurement units where present
  • Nomenclature table: “representance” -> “significance”
  • Line 24: please indicate the measurement units in the form of power (e.g., m2/h -> m2·h-1)
  • 2.1: sixth line “0.4 m/s” -> “0.4 m·s-1
  • 2.1: last line “r/min” -> “rpm” or “rev·min-1
  • Line 179: “kpa” -> “kPa”
  • Line 185: “MN/m2” -> “MN·m-2
  • Line 189: “The total weight of this part of the soil was about 47 kg.” -> “The total mass of this part of the soil was about 47 kg.” (pay attention to the difference between weight and mass)
  • Lines 241-380: why exposing the DEM simulations before the fluid dynamic simulations with Fluent on the air duct? This is still a point to be fixed. The article should be heavily rearranged and the connections between subsequent paragraphs evidenced more. I think that a correct way of operating is using the fields of motion of the air (from the Fluent simulations) as a starting point to simulate the particle removal on the soil ridge, and then verify in the field on a real prototype.
  • Lines 241-380: please simulate also the scenario in which the vine plants (maybe with a standardized shape) are buried under the ridge, and compare the results with the scenario without vine plants (the only one present at now), to demonstrate the influence (or not) in retaining the soil.
  • Line 248: what is “cfm”?
  • Line 319: “50 m/s” -> “50 m·s-1
  • Lines 127, 328, 621: put the space between number and unit
  • Tab 3.1 header “m/s” -> “m·s-1
  • Headers of all the tables: please indicate the measurement unit of each column between square brackets, e.g. “Soil ridge mass at 0.43 s /kg” -> “Soil ridge mass at 0.43 s [kg]”
  • Throughout the article: composite measurement units require always the multiplicative dot, e.g. “ms-1” -> “m·s-1
  • Line 15, 16 (abstract) Pay attention that contracted forms (e.g. “isn’t”, “couldn’t”) are not used in formal English (i.e. the English that should be adopted for an article) and, therefore, should be avoided
  • Table 2.1: header “Technical parameters of the machine” -> “Technical parameter of the machine”
  • Table 2.1: please could you add some parameters more (e.g. the fan impeller diameter, the air duct section at the end of the airduct, the air speed range)? You should put the reader in the condition of being able to replicate your experiments. Furthermore, as I specified in my previous review, please add to this table also the parameters that were object of study/optimization, clearly underlining this and indicating also the investigated values or range of values (in the simulations or in the field experimentation).
  • Lines 101-102 “The soil cleaner machine included three major components: a chassis, a fan and a power device, as shown in Figure 2.1.” -> “As shown in Figure 2.1, the soil cleaner machine included three major components: a chassis (in grey), a fan (in blue) and a power device (in green).”
  • Paragraph 2.3.1.1/2/3 (Lines 156-202): where are indicated the international norms followed for the procedure of soil moisture determination, ring-knife method, elastic modulus calculation, static friction angle calculation?
  • 2.2 and 2.4: please insert the nu symbol (ν)
  • 2.6: according to the used measurement units, the result has the dimension of “m3·s-1”, not only “m3”. Please correct.
  • Line 219: “Where μ=0.2.” -> “Where μ=0.2 is the static friction coefficient between soil particles”.
  • Lines 213-219: as far as I see, you used 0.2 as static friction coefficient; how did you calculate it (or in which reference did you find it)? It seems too low (cfr: rubber-on-ice 0.15, rubber-on-wet-concrete 0.3). If you only referred to the repose angle of sand, the friction coefficient would be 0.7. Please check the coefficient and do again all calculations. Also (again in this revision round), how did you consider the moisture level in determining the fore to be overcome by the air stream? Try to give a look to the literature: moisture can raise significantly the coefficient of friction (it triggers a sort of adhesion effect). As you check the moisture content of the soil, how did you use this datum?
  • Lines 265-267: the copper-aluminium selection procedure is not clear at all to a reader, you have to spend some words more, firstly reporting the resistivity of copper and of aluminium is 65×10−8 Ω·m / 1.68×10−8 Ω·m and their ratio, 1.58.
  • Then in Eq. 2.14 you have calculated the current in a wire starting from the power and the voltage, but the material properties do not intervene (the formula is general: P = V I -> I = P/V), so what you obtain is the same regardless the material. In Eq. 2.13 you simply multiply by 1.5 the result of the previous calculation but the result is still a current so it is not clear what you have done (and maybe not correct). Please revise completely all this paragraph and refer to the Ohm’s laws to calculate the section of metal required in the two scenarios and then expose the reader your choice.
  • Lines 372-373: you have declared that you did not modify the height of the air outlet when changing the angle. But doing so, the amount of material invested by the airstream change a lot (the results vary also due to the ridge area interested by the wind stream, you are not only testing the incidence angle), unless something in the geometry could be not clear to me (I suggest to add a scheme of the situation see from a lateral point of view).
  • Line 508: “and the greater the dynamic pressure.” -> “and the greater the dynamic pressure obtained on the target external surface, i.e. on the soil ridge” (correct as indicated just to avoid any possible misunderstanding with the static pressure on the pipe walls)
  • Lines 568-590: I have still many concerns concerning the design of this experimental part, and the authors have not answered to my observations, so I propose them again hereinafter. The way the field experimentation was led rises many problematic points: according to what was reported in the text, the experimentation seems to be almost useless, because many parameters were out of the control of the experimenters and, also, no metric was set to quantify the progress of the soil removal from the ridge during the operation of the machine. My suggestion is to repeat the experimentation and fill the important gaps highlighted above. About the progressive quantification of the machine effect, I suggest, for example, to use a visual support (e.g. photographs) to measure the ridge profile in set time intervals and quantify ex post the volume (hence the mass) of removed soil. The best way would be put all the terrain (and the vine plants) on a scale and measure with continuity the mass decrement over the time.

Author Response

请参阅该文件。

Round 3

Reviewer 2 Report

  • Line 199: if you say “The volume V and mass m of the soil that the soil cleaner need to remove per second” you are referring to a volumetric rate and a mass rate, so please indicate them as dV/dt and dm/dt and express the results in m3s-1 and kg·s-1.
  • Line 207: please insert a reference to justify the assumption of the friction coefficient value.
  • Line 216: the fan indicated this time is 9-26 22 kW. Previously was 4-72 15 kW. Rotational speed, weight, dimensions are obviously different. It seems you did not use any of these calculations to really choose your fan (and other components), you are only trying to match it ex post. If this is the case, you have to change all your work, presenting it as it is, i.e. a study on an existing machine (built by you or not, this is not important), not the study for the design and the dimensioning of a machine that you have developed.
  • Line 224: you wrote “The diesel generator with a rated power of 50 kW was selected based on the generator efficiency of 20%.” Have you measured this efficiency? Was it written somewhere? By making a quick calculation, with that indicated efficiency I will obtain 10 kW at the generator output, so less than the 22 kW indicated by you for the fan. Did you really use this data?
  • Line 226-228: the cable selection process is completely opaque for the reader; which is the “selection principle of electrical cables”? I was not able to find the indicated reference, please indicate the DOI (or an URL). How long are the cables? 30 m (line 346)? You should report the datum also here. If you used one of the many cable-dimensioning-calculator in the Internet, you have also to put the reader in a position to validate your calculations and, in case, to adapt them to his cases, so you have to indicate the parameters used for the calculation (e.g., the maximum voltage drop, the type of insulation, the cable installation mode) and the URL of the calculator.
  • Paragraph 2.4: according to your “new” data, the wind speed now is 68 m·s-1 but the pictures are the same of the previous version (v=50)…
  • Line 318: pay attention, there is a label in Chinese, translate it in English.
  • Line 327: “The simulation of simulated grape vines in the soil ridge was as follows” -> “The simulation process of ground removal with grape vines inside the soil ridge was reported hereinafter”.
  • Line 328: please give geometrical dimensions of the modelled grape vines.
  • After line 331: please insert a paragraph in which you compare the results without/with modelled grape vines, arriving to a table similar to Tab. 2.5 and maybe quantifying the influence of grape vines.
  • Line 354: you are referring to 15 kW, it should be 22 kW (latest version)
  • Line 434: “the soil quality decreased slowly within” -> “the soil quantity decreased slowly within” (check also line 436)
  • 3.3 caption: “Soil clearing effect” -> “Soil cleaning effect”
  • Paragraph 3.5.2.: It is not sufficient to show only that your machine works but also that the results are lined up with your calculations, otherwise they could seems not so useful from the point of view of the power-sizing (even because the numbers have changed a lot through these versions and some passages that lead to the building of the machine are not yet so explicit); ok that the season is not the right one, but I strongly suggest you to setup an indoor experiment with your machine, by recreating a stretch of ridge of the same dimensions of the ones on the field and by using your machine to then demolish it. In the meantime take photos at regular intervals, or put a scale under your mock-up ridge to have a continuous mass measurement.
